# Enhancing Sustainable Finance through Green Hydrogen Equity Investments: A Multifaceted Risk-Return Analysis

Cristiana Tudor 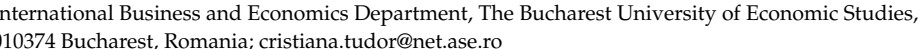

International Business and Economics Department, The Bucharest University of Economic Studies, 010374 Bucharest, Romania; cristiana.tudor@net.ase.ro

**Abstract:** Amidst the global push for decarbonization, green hydrogen has gained recognition as a versatile and clean energy carrier, prompting the financial sector to introduce specialized investment instruments like Green Hydrogen Exchange-Traded Funds (ETFs). Despite the nascent nature of research on green hydrogen portfolio performance, this study examines two key green hydrogen ETFs (i.e., HJEN and HDRO) from April 2021–May 2023, aiming at conducting a multifaceted exploration of their performance, isolating and measuring their sensitivity to the primary market factor, and assessing the capabilities of systematic trading strategies to preserve capital and minimize losses during market downturns. The results spotlight lower returns and higher risks in green hydrogen investments compared to conventional equity (proxied by ETFs offering exposure to developed markets—EFA and emerging markets—EEM) and green energy portfolios (proxied by the ETF ICLN). To comprehensively evaluate performance, an array of risk-adjusted metrics, including Std Sharpe, ES Sharpe, VaR Sharpe, Information ratio, Sortino ratio, Treynor ratio, and various downside risk metrics (historical VaR, modified VaR, Expected Shortfall, loss deviation, downside deviation, and maximum drawdown) are employed, offering a nuanced understanding of the investment landscape. Moreover, single-factor models highlight significant systematic market risk, reflected in notably high beta coefficients, negative alphas, and active premia, underscoring the sensitivity of green hydrogen investments to market fluctuations. Despite these challenges, a silver lining emerges as the study demonstrates the efficacy of implementing straightforward Dual Moving Average Crossover (DMAC) trading strategies. These strategies significantly enhance the risk-return profile of green hydrogen portfolios, offering investors a pathway to align financial and social objectives within their equity portfolios. This research is motivated by the need to provide market players, policymakers, and stakeholders with valuable insights into the benefits and risks associated with green hydrogen investment, considering its potential to reshape the global energy landscape.

**Keywords:** green hydrogen equity; downside risk; market risk; risk-adjusted returns; factor models; DMAC

## 1. Introduction

The paradigm of socially responsible investing (SRI) represents a fundamental shift in the way investments are evaluated and selected, integrating environmental, social, and governance (ESG) criteria into traditional financial analysis (Vo et al. 2019). This approach has gained substantial momentum during the most recent decades (Renneboog et al. 2008; Junkus and Berry 2010; Ito et al. 2013; Brzeszczyński and McIntosh 2014), reflecting a growing awareness of the interconnectedness of financial markets and the broader societal and environmental contexts in which they operate (Kiesel and Lücke 2019; Tripathi and Kaur 2020; Camilleri 2021; Driessen 2021).

However, the literature has provided mixed evidence supporting the merits of SRI from a risk-return perspective. Kempf and Osthoff (2007), among others, show that a self-constructed high-SRI portfolio performs better than a low-SRI portfolio in terms of abnormal returns. Additionally, Brzeszczyński and McIntosh (2014) establish the positive per-

formance of British SRI stocks during the 2000–2010 period, as evidenced by risk-adjusted measures such as the modified Sharpe ratio (MSR) and certainty equivalent (CEQ) returns. Wu et al. (2017) reinforce these findings by comparatively studying the performance of two FTSE value-weighted investment portfolios, i.e., a SRI portfolio and a conventional portfolio, from 2004 to 2011. However, a meta-analysis of 85 studies and 190 experiments performed by Revelli and Viviani (2015) concludes that considering corporate social responsibility in equity portfolios is neither a weakness nor a strength compared with conventional investments. Furthermore, other studies encounter no relationship between socially responsible investment (SRI) and portfolio performance (Zehir and Aybars 2020). These findings underscore the need for additional evidence aimed at investigating whether screening for responsibility impacts portfolio performance and whether financial goals and social objectives can be concomitantly pursued in equity portfolios.

Green hydrogen, produced using the process of water electrolysis powered by renewable energy sources, represents a promising solution in the quest to decarbonize the global economy (Zerta et al. 2008; Kovač et al. 2021; Oliveira et al. 2021; Kumar and Lim 2022; Squadrito et al. 2023) with the International Energy Agency (IEA) establishing its key role for the world's clean energy future (Biggins et al. 2022). Unsurprisingly, statistics indicate that the global hydrogen market is set to grow from 70 million tonnes in 2019 to 120 million tonnes by 2024 (Safari and Dincer 2020; Atilhan et al. 2021; Osman et al. 2022). Its potential to serve as a clean and versatile energy carrier, along with its capacity to mitigate carbon emissions, has additionally spurred substantial interest among investors. As a testament to this growing enthusiasm, the financial sector has introduced specialized investment vehicles, such as Green Hydrogen Exchange-Traded Funds (ETFs), designed to offer exposure to this nascent sector. Thus, amidst the proliferation of green energy investment options, Green Hydrogen ETFs have recently emerged to provide investors with exposure to the green hydrogen sector. As the literature suggests, ETFs have gained prominence for their ease of access, diversification benefits, and transparency. However, research on the performance of Green Hydrogen ETFs is still in its infancy. This research paper fills a crucial gap by conducting a comparative analysis of the risk-return characteristics of Green Hydrogen ETFs, conventional equity portfolios, and broader green energy portfolios, utilizing a plethora of relevant tools. Particularly, this study paper delves into the performance of Green Hydrogen ETFs, comparing their risk-return characteristics with conventional equity portfolios and green energy portfolios. The comparative perspective of this research is especially significant as it not only assesses Green Hydrogen ETFs against conventional equity portfolios but also positions them within the context of existing green energy investments. This triangulation allows for a nuanced understanding of how green hydrogen ETFs perform in relation to different investment options, enabling stakeholders to make more informed decisions in their pursuit of sustainable and profitable investment strategies.

Of note, the period under investigation spans after the first year of the COVID-19 pandemic, which has left an indelible mark on the energy market, prompting a reassessment of risk and return dynamics within the realm of Energy Exchange-Traded Funds (ETFs) (IEA 2020). The unprecedented global health crisis led to lockdowns, travel restrictions, and economic uncertainties (Ibn-Mohammed et al. 2021; Khan et al. 2022; Tudor 2022a, 2022b), resulting in a substantial reduction in greenhouse gas (GHG) emissions due to decreased industrial activities and reduced travel (Le Quéré et al. 2020; Sharma et al. 2020; Tudor 2022a). The widespread lockdowns and economic slowdowns (Tudor 2022c) additionally resulted in a significant drop in energy demand, causing disruptions across various energy sources (Jiang et al. 2021; Hoang et al. 2021). While the initial impact led to a decline in carbon emissions due to reduced industrial activity and transportation, the renewable energy sector faced both challenges and opportunities (Eroğlu 2021). On the one hand, delays in project timelines and disruptions in the supply chain posed obstacles to renewable energy deployment (Olabi et al. 2022). On the other hand, the post-pandemic stimulus packages and an increase in the number of renewable energy incentives heightened the focus on green energy investments in the aftermath of the COVID-19 pandemic (Pradhan

et al. 2020; Hoang et al. 2021). This unique context adds layers of complexity to the risk-return analysis of energy ETFs during the last pandemic waves and the post-pandemic recovery period spanning from April 2021 to May 2023.

The motivation behind this analysis is to provide investors, policymakers, and stakeholders with valuable insights into the potential benefits and risks associated with green hydrogen ETFs. Given the substantial market growth and heightened enthusiasm surrounding green hydrogen, a comprehensive evaluation of the financial performance of green hydrogen investments is both timely and pivotal.

The main goals of the research paper can be described as follows: (i) To conduct a multifaceted exploration of the performance of green hydrogen portfolio investments within the context of the broader financial landscape in the post-pandemic era; (ii) To isolate and measure the sensitivity of Green Hydrogen ETFs to the primary market factor; (iii) To assess whether systematic trading strategies can serve as effective tools for preserving capital and minimizing losses in the green hydrogen sector. In order to attain its goals, the study pursues several distinct directions with notable contributions. First, it embarks on a multifaceted exploration of the performance of Green Hydrogen ETFs and compares it to two conventional equity portfolios (a developed market portfolio and an emerging market portfolio) and a broader green energy portfolio using a range of risk-adjusted performance metrics, including the standard Sharpe ratio, Expected Shortfall (ES) Sharpe, Value-at-Risk (VaR) Sharpe, and the Sortino ratio, thus providing a comprehensive evaluation of portfolio performance that goes beyond traditional measures. Furthermore, it delves into the vital aspect of downside risk-return trade-offs by employing various downside risk statistics (i.e., conventional and modified versions of Value-at-Risk (VaR) and Expected Shortfall (ES), as well as downside deviation and maximum drawdown), which is pivotal for investors navigating the turbulent waters of financial markets. By considering these different ratios in tandem, investors gain a more holistic view of a portfolio's risk-return profile, allowing for a more nuanced assessment of the trade-offs between risk and return. This multi-dimensional approach is particularly valuable in volatile markets or when downside protection is of paramount importance, as it enables investors to make more informed decisions tailored to their specific risk preferences and investment objectives. Third, it employs single-factor models to elucidate the return variability of two Green Hydrogen Exchange-Traded Funds (ETFs), which is crucial for understanding the specific factors driving the performance of these investment instruments. By isolating and measuring the sensitivity of Green Hydrogen ETFs to the primary market factor, investors can gain valuable insights into the extent to which these funds are exposed to systematic market fluctuations. These models not only help quantify the risk associated with investing in Green Hydrogen ETFs but also contribute to a more comprehensive comprehension of how these investments behave within the context of the broader financial landscape. Finally, it offers compelling evidence that DMAC strategies are effective tools for preserving capital and minimizing losses in the green hydrogen sector. By optimizing entry and exit points based on moving average crossovers, the study shows that systematic strategies not only protect investments during market downturns but also pave the way for more resilient and potentially more profitable investment approaches, thus allowing investors to stay agile and responsive to the evolving landscape of green hydrogen, providing a critical edge in this ever-changing market.

The results of this study show that green hydrogen investments present lower return performance and higher risk than both conventional equity and green energy portfolios during April 2021–May 2023. Moreover, single-factor models exhibit noteworthy explanatory power for the two green hydrogen portfolios, demonstrating that a significant proportion of the ETFs' returns can be attributed to systematic market risk. Furthermore, the study presents compelling evidence of the effectiveness of Dual Moving Average Crossover (DMAC) strategies in the green hydrogen sector, providing investors with resilience and potentially more profitable investment approaches.

The rest of the paper is structured as follows: Section 2 presents and discusses the main results; Section 3 contains the methodology, including trading strategies and the data used in the paper; and Section 4 concludes.

## 2. Results and Discussion

Figure 1 plots the evolution over time of the daily closing prices for the five selected ETFs.

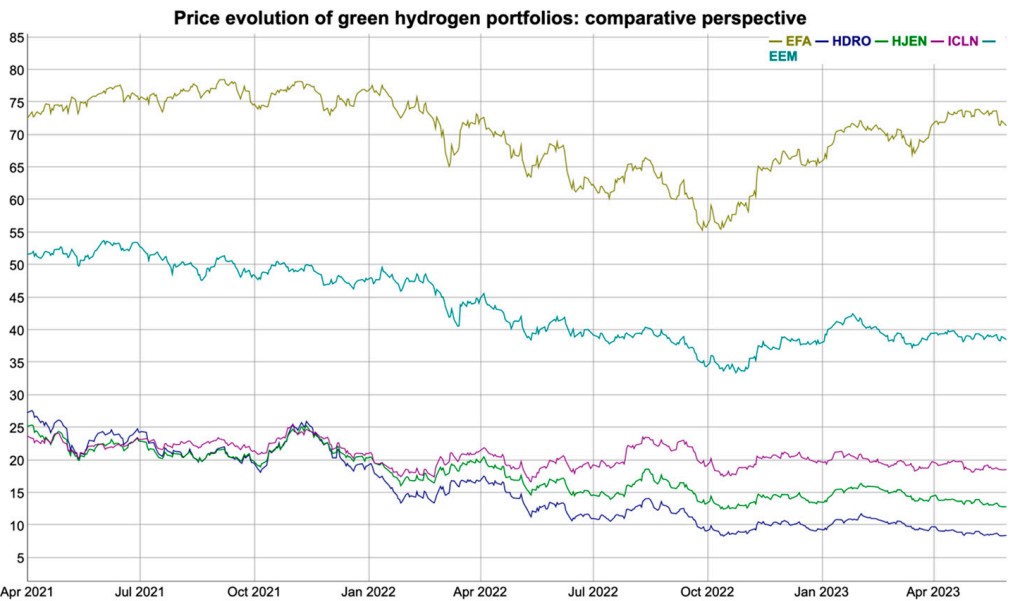

**Figure 1.** Price evolution of the five investment funds (1 April 2021–31 May 2023).

The price evolution chart of these five ETFs from April 2021 to May 2023 presents an intriguing narrative of changing market dynamics. During the initial period, from April 2021 to October 2022, all five ETFs exhibited a consistent overall downturn, indicative of broader market challenges or specific sector headwinds affecting their performance. Factors such as global economic uncertainties, geopolitical events, or industry-specific challenges could have contributed to this synchronized decline. However, starting in October 2022 and continuing until May 2023, a noteworthy divergence emerged. The Developed Markets ETF (EFA) embarked on an upward trajectory that starkly contrasts with the other four ETFs (EEM, HJEN, HDRO, and ICLN), which either showed a less accentuated upward movement (EEM) or remained in a downtrend (ICLN, HJEN, and HDRO). This divergence implies a decoupling of performance drivers, with EFA potentially benefiting from factors or conditions not shared by the other ETFs. As such, the discrepancy may be attributed to several factors. One plausible explanation is that investors began to favor developed markets over emerging markets and green energy sector-focused ETFs. Improved economic conditions and more stable regulatory environments in developed markets, coupled with constant or reduced geopolitical tensions and the subdue of recession-related effects by the end of 2022, have sparked renewed interest in developed market equity investments. Additionally, the developed market portfolio may have been less exposed to particular risk factors affecting the other ETFs. This divergence further underscores the significance of monitoring changing market dynamics and the value of diversification. It also highlights the need for investors to remain adaptable and responsive to evolving trends in the financial landscape.

The risk/return scatter chart (Figure 2) depicting the annualized performance of the five portfolios (EFA, EEM, HJEN, HDRO, and ICLN) from April 2021 to May 2023 reinforces the above-mentioned preliminary remarks by providing valuable insights into their comparative risk-return profiles.

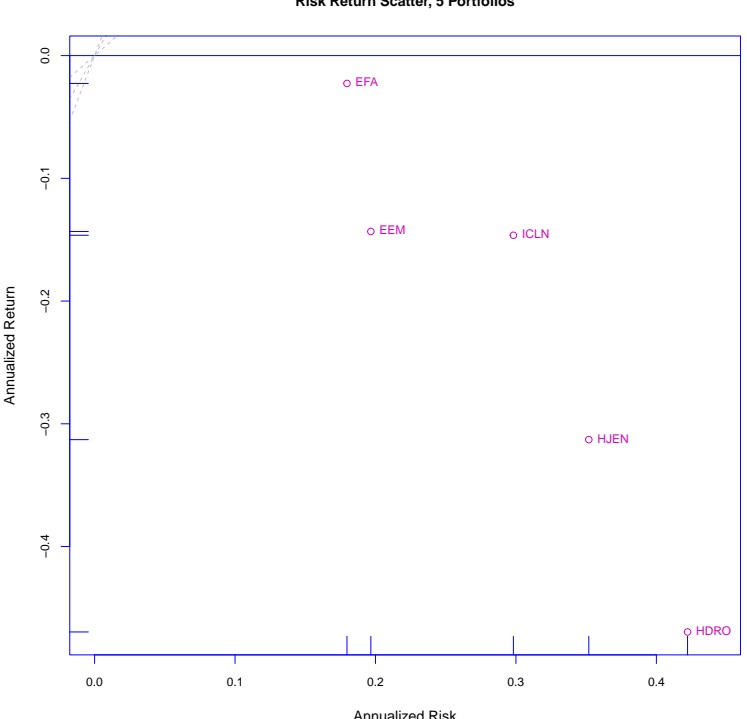

**Figure 2.** The risk/return scatter chart of the five investment funds over 1 April 2021–31 May 2023.

EFA, representing developed markets, displays the most favorable risk-return profile, with the highest return and lowest risk among the portfolios. This aligns with the historical perception of developed markets as offering greater stability, although it is notable that all the portfolios exhibited negative returns during this period. The negative returns across the board reflect the complex and volatile market conditions that have characterized the post-pandemic era. The other portfolios, including EEM, HJEN, HDRO, and ICLN, exhibit lower performance, with all green-energy portfolios, particularly HDRO, demonstrating the most significant underperformance. These results underscore the need for diversified portfolios and performant strategies that can weather challenging market conditions, supporting Kiani's conclusion (Kiani 2011). Additionally, they highlight the importance of rigorous risk management strategies and the value of monitoring market trends that can produce a substantial impact on investment outcomes, especially in specialized sectors like hydrogen or clean energy (Marti-Ballester 2019). The negative returns observed across the portfolios reflect a period of economic uncertainty and structural changes in global financial markets. Moreover, the dispersion in performance across the five portfolios underscores the importance of thorough portfolio analysis and the role of market dynamics in shaping investment outcomes.

Figure 3 next depicts the cumulative returns for EFA, HJEN, and HDRO over the period from April 2021 to May 2023 and provides a visual snapshot of the performance disparities among these assets.

Notably, the chart shows that the value of a one-dollar investment in EFA, representing the broader developed market index, remains relatively stable, closing near one by the end of the period. In contrast, the cumulative returns for the two green hydrogen ETFs, HJEN and HDRO, are notably lower, ending slightly above 0.4 and around 0.35, respectively. The cumulative return for EFA suggests a more stable performance, reflecting the relative maturity and resilience of developed market investments. Conversely, the lower cumulative returns for HJEN and HDRO indicate a more challenging path, likely influenced by the specialized and evolving nature of the green hydrogen sector.

The rolling 120-day performance chart (Figure 4) also highlights the divergent performance paths of traditional equity investments and green hydrogen ETFs.

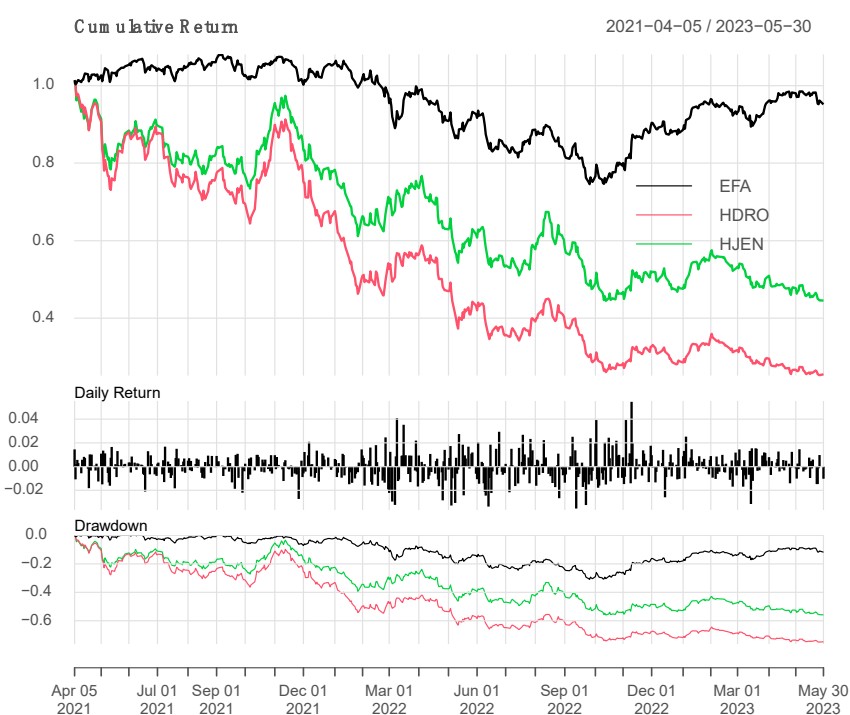

**Figure 3.** Equity curves of the two green hydrogen ETFs and EFA over 1 April 2021–31 May 2023.

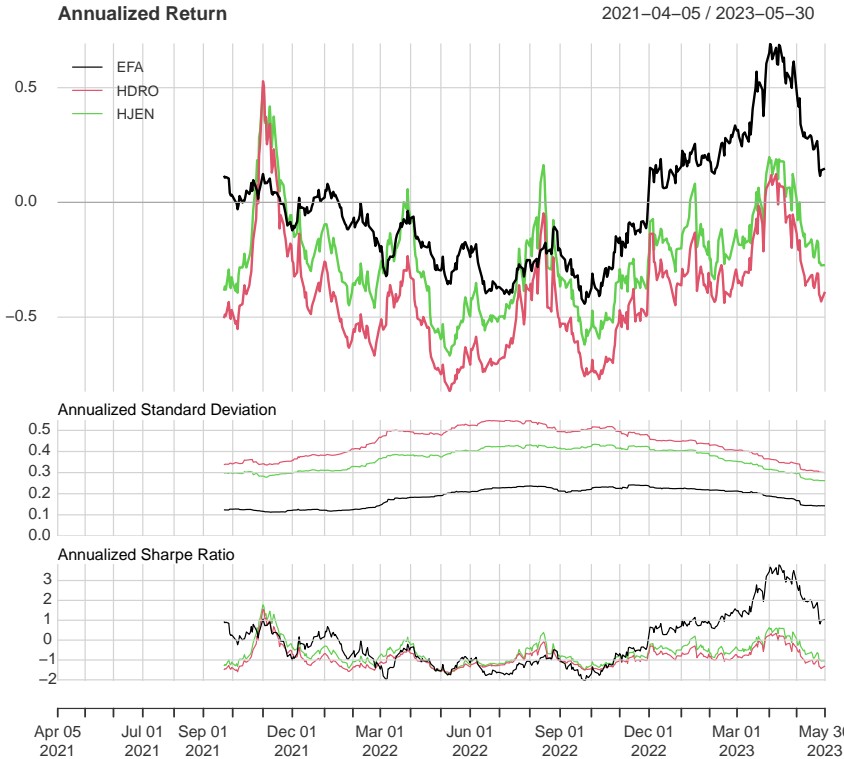

**Figure 4.** Rolling 120-day performance of the two green hydrogen ETFs and EFA over 1 April 2021–31 May 2023.

The annualized rolling return for EFA consistently surpasses that of the two green hydrogen ETFs and remains positive after 2022, indicating a more stable and consistently favorable performance. The two green hydrogen investments exhibit both lower and more volatile returns, which in turn contribute to their significantly lower overall risk-adjusted performance, as evidenced by the lower annualized rolling Sharpe ratios.

The risk-adjusted performance metrics and downside risk statistics (Table 1) for the five portfolios (EFA, EEM, HJEN, HDRO, and ICLN) over the period from April 2021 to May 2023 offer further perspective on their performance.

**Table 1.** Risk-adjusted performance metrics and downside risk statistics.

| | EFA | HDRO | HJEN | ICLN | EEM |
|---|---|---|---|---|---|
| StdDev Sharpe (Rf = 0%, $p$ = 95%): | −0.002402 | −0.08127 | −0.05607 | −0.02409 | −0.04335 |
| VaR Sharpe (Rf = 0%, $p$ = 95%): | −0.00154 | −0.05064 | −0.03436 | −0.01581 | −0.02877 |
| ES Sharpe (Rf = 0%, $p$ = 95%): | −0.00115 | −0.04133 | −0.02754 | −0.01272 | −0.02293 |
| Sortino Ratio (MAR = 0%) | −0.003434 | −0.11296 | −0.07832 | −0.03511 | −0.0608 |
| Downside deviation | 0.0079 | 0.0191 | 0.0159 | 0.0129 | 0.0088 |
| Maximum drawdown | 0.3094 | 0.7501 | 0.5591 | 0.3652 | 0.3965 |
| Historical VaR | −0.0171 | −0.0416 | −0.0351 | −0.0291 | −0.0194 |
| Modified VaR | −0.0176 | −0.0427 | −0.0362 | −0.0286 | −0.0187 |
| Historical ES | −0.0247 | −0.0529 | −0.0446 | −0.0374 | −0.0262 |
| Modified ES | −0.0235 | −0.0523 | −0.0451 | −0.0356 | −0.0234 |

EFA, representing developed markets, emerges as the top performer in terms of risk-adjusted return and downside risk protection. This is supported by the highest values across various risk-adjusted metrics, such as the Sharpe ratio, VaR Sharpe, ES Sharpe, and Sortino ratio, all indicative of the portfolio's superior risk-adjusted returns. EEM, focusing on emerging markets, follows closely, highlighting the potential for competitive performance in this category. Notably, the two green hydrogen ETFs, HJEN and HDRO, exhibit notably lower risk-adjusted returns and higher downside risk metrics, underlining the elevated risk associated with sector-specific investments. As such, focused investments, such as green hydrogen ETFs, come with potential opportunities but demand a closer look at risk management and potential downside scenarios. The significant maximum drawdowns of 0.75 and 0.55 observed in the two green hydrogen ETFs over the period from April 2021 to May 2023 indicate the extent to which the values of these ETFs have fallen from their peak levels during the given timeframe, signifying the magnitude of losses that investors may have encountered. In contrast, the maximum drawdowns of 0.30 and 0.39 for EFA and EEM, representing the broader market indices, are notably lower.

The Snail Trail chart (Figure 5) depicting the risk versus return for HDRO and HJEN over 120-day rolling periods from April 2021 to May 2023 also reveals the persistent challenge faced by these green hydrogen ETFs.

Notably, the chart illustrates that the return for both HDRO and HJEN remains consistently in negative territory throughout the entire period. This observation highlights the prolonged underperformance of green hydrogen ETFs in a turbulent market environment characterized by economic uncertainties, supply chain disruptions, and regulatory changes. Such extended periods of negative returns indicate the difficulty in achieving profitability and stable performance within the green hydrogen sector during this specific timeframe. It underscores the need for investors and asset managers to exercise caution and consider risk management strategies when investing in highly specialized sectors like green hydrogen, which can be more susceptible to market volatility.

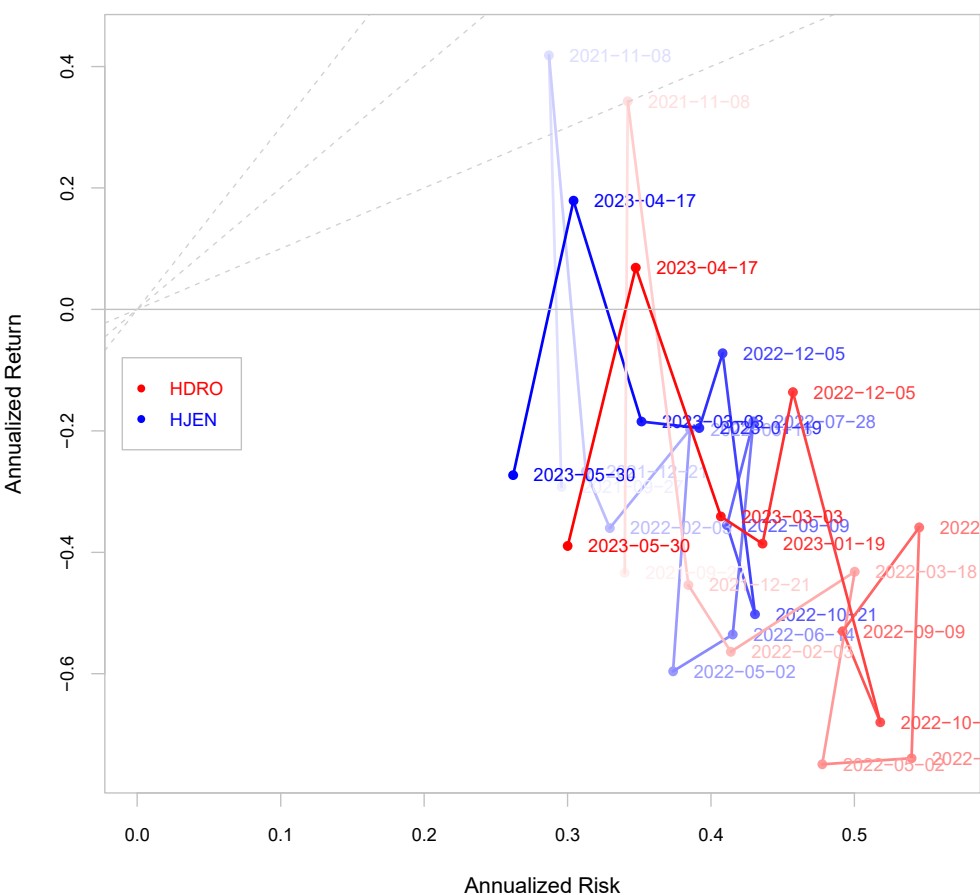

**Figure 5.** SnailTrail chart for the two green hydrogen ETFs over 1 April 2021–31 May 2023.

Table 2 reports the results of evaluating the single-factor model for the four portfolios (EEM, HJEN, HDRO, and ICLN), with EFA representing the market index, over the period from April 2021 to May 2023, and contributes to presenting a comprehensive view of their risk exposures and performance characteristics.

**Table 2.** Single-factor model results.

|  | **HDRO to EFA** | **HJEN to EFA** | **ICLN to EFA** | **EEM to EFA** |
|---|---|---|---|---|
| Alpha | −0.0021 | −0.0012 | −0.0000 | −0.0000 |
| Beta | 1.5106 | 1.3021 | 0.9734 | 0.8553 |
| R-squared | 0.4137 | 0.4424 | 0.3443 | 0.6107 |
| Annualized Alpha | −0.4143 | −0.2625 | −0.1018 | −0.1215 |
| Correlation (*p*-value) | 0.6432 (0) | 0.6651 (0) | 0.5868 (0) | 0.7815 (0) |
| Active premium | −0.4469 | −0.2902 | −0.1237 | −0.1206 |
| Information ratio | −1.33 | −1.0814 | −0.5123 | −0.9614 |
| Treynor ratio | −0.3109 | −0.2403 | −0.1504 | −0.1676 |

It is observed that EEM and ICLN exhibit subunitary betas, indicating that their returns are less sensitive to market movements compared to the broader market represented by EFA. This aligns with the idea that emerging markets (EEM) and green energy (ICLN) may possess different risk profiles compared to the broader market. In contrast, HJEN and HDRO display high betas (1.3 and 1.5), suggesting a stronger response to market movements, which may be associated with the heightened volatility in the green hydrogen sector.

All portfolios exhibit negative alfas, active premia, information ratios, and Treynor ratios, indicating underperformance relative to the single-factor model. The two green hydrogen ETFs (HJEN and HDRO) stand out for having the lowest values, while EEM and ICLN exhibit somewhat higher, albeit still negative, metrics. The negative values are suggestive of challenges in generating excess returns compared to the expected return based on the market factor. This could be attributed to the structural and market-specific factors influencing these portfolios. These findings align with Marti-Ballester (2019), suggesting that investment in clean energy funds carries a financial cost for investors in comparison to conventional fund investments.

The correlation coefficients reveal that EEM has the highest correlation with the market (0.7), reflecting its responsiveness to broader market movements. ICLN, representing green energy, exhibits the lowest correlation (0.5), indicating a level of independence from the overall market. The green hydrogen ETFs (HJEN and HDRO) fall in between, suggesting a moderate degree of correlation with market dynamics. Figure 6 provides a visual representation of these links.

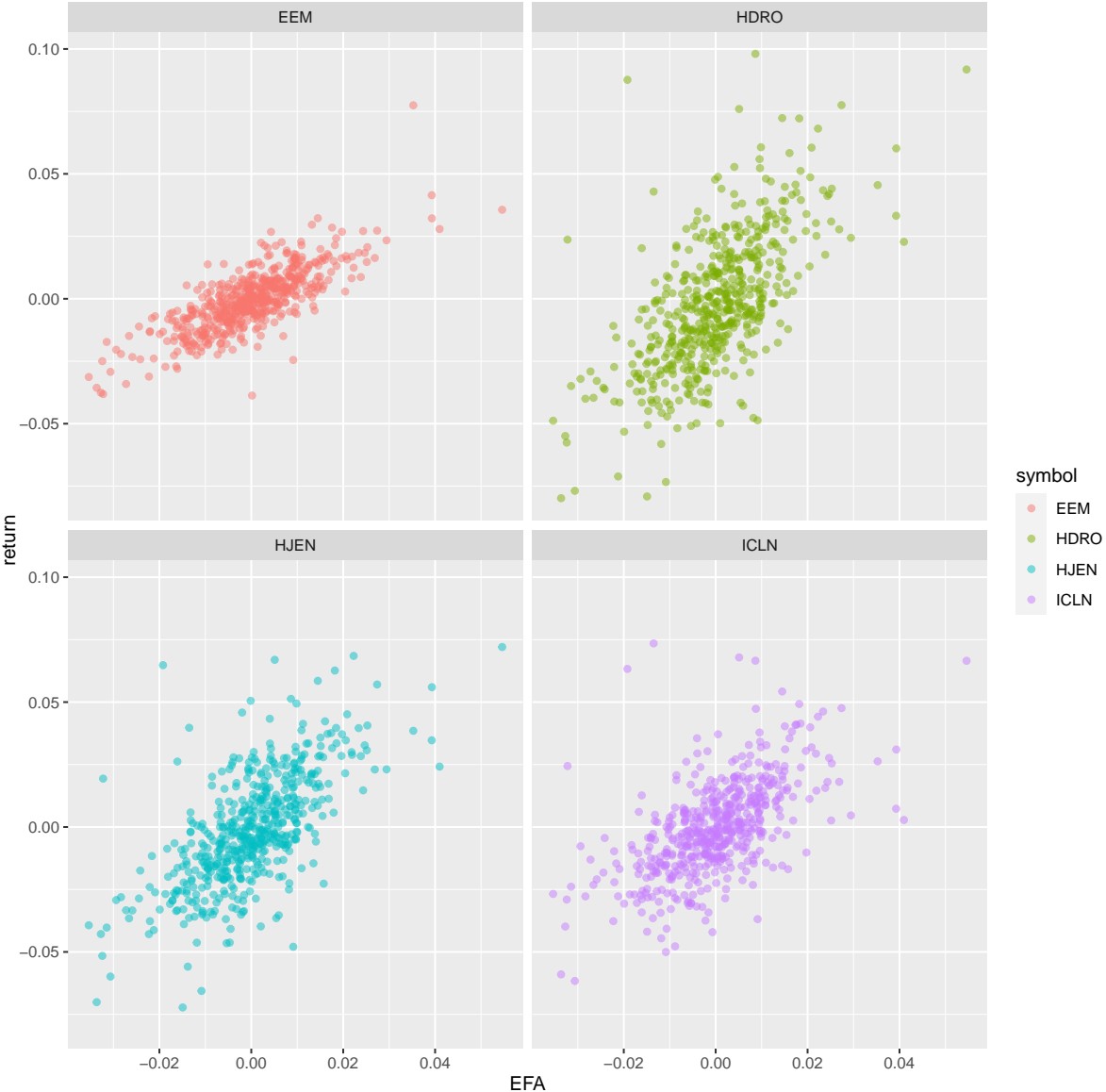

**Figure 6.** Relationship between energy ETFs return and the market index (EFA) over 1 April 2021–31 May 2023.

Moreover, the R-squared values mirror these trends, with EEM displaying the highest, ICLN the lowest (0.34), and the green hydrogen ETFs intermediate R-squared values. In particular, the single-factor model's ability to explain over 40% of return variability for two green hydrogen ETFs, HJEN and HDRO, as evidenced by R-squared values of 41% and 44%, offers a noteworthy insight into the factors influencing the performance of these specialized investment vehicles. The relatively high R-squared values suggest that a significant portion of the variability in returns can be attributed to the market factor represented by the broader market index (EFA). This finding aligns with the idea that green hydrogen ETFs remain sensitive to market dynamics, even though they represent a niche sector within the broader investment landscape. The R-squared values also indicate that while sector-specific factors play a role in return variability, market-wide movements, and systemic risk factors still exert a substantial influence on the performance of these ETFs.

Of note, the decision to rely on single-factor models, particularly the Capital Asset Pricing Model (CAPM), was driven by the unique characteristics of the volatile green energy space, especially within the context of green hydrogen. One of the primary aims of the research was to provide investors with a clear and straightforward analysis of the sensitivity of green energy ETFs to the primary market factor. In this context, the simplicity of the CAPM allows the isolation and measurement of systematic risk, which is crucial for understanding the fundamental forces influencing these investments.

All analyses suggest that green hydrogen ETFs face challenges in rewarding investors due to sector-specific headwinds and market uncertainties, in line with previous research (i.e., Jones et al. 2008).

The implications of the observed trends within the broader financial context are multifaceted. The upward trajectory of the Developed Markets ETF (EFA) may signal a recovery in developed economies, potentially influenced by improved economic indicators, policy initiatives, or other positive factors. The less pronounced recovery in the Emerging Markets ETF (EEM) may reflect a more cautious investor sentiment or persistent challenges in emerging economies.

In the green energy sector, the divergence among ICLN, HJEN, and HDRO raises questions about the specific challenges facing these subsectors. Factors such as regulatory changes, technological advancements, or shifts in public sentiment toward different forms of green energy could be influencing their performance differentially.

Overall, this narrative underscores the importance of monitoring changing market dynamics and recognizing the nuanced performance of various sectors within the broader financial landscape. Investors and analysts should delve deeper into the specific factors impacting green energy ETFs to make informed decisions and capitalize on emerging opportunities or navigate challenges in this evolving market.

In this context, the importance of implementing trading strategies capable of improving performance and minimizing risks in green hydrogen investments cannot be overstated, particularly in a sector as dynamic and susceptible to volatility as the green hydrogen industry. As such, the unique challenges and opportunities in this sector demand a proactive approach to portfolio management, whereas trading strategies tailored to the specific characteristics of green hydrogen investments can help investors navigate the complexities of the market. Particularly, DMAC strategies have been widely recognized for their ability to capture trends and generate buy and sell signals based on moving average crossovers. In the context of green hydrogen investments, where market dynamics can be intricate and subject to rapid shifts, this strategy serves as a valuable tool for investors.

Figures 7 and 8 illustrate the Simple Moving Average (SMA) 20 versus SMA 120 crossovers for HJEN and HDRO and reveal the occurrences of crossovers over the analysis period, where the shorter-term SMA 20 crosses above or below the longer-term SMA 120, serving as potential buy or sell signals. The charts also map (in red) the standard deviation around the longer moving average.

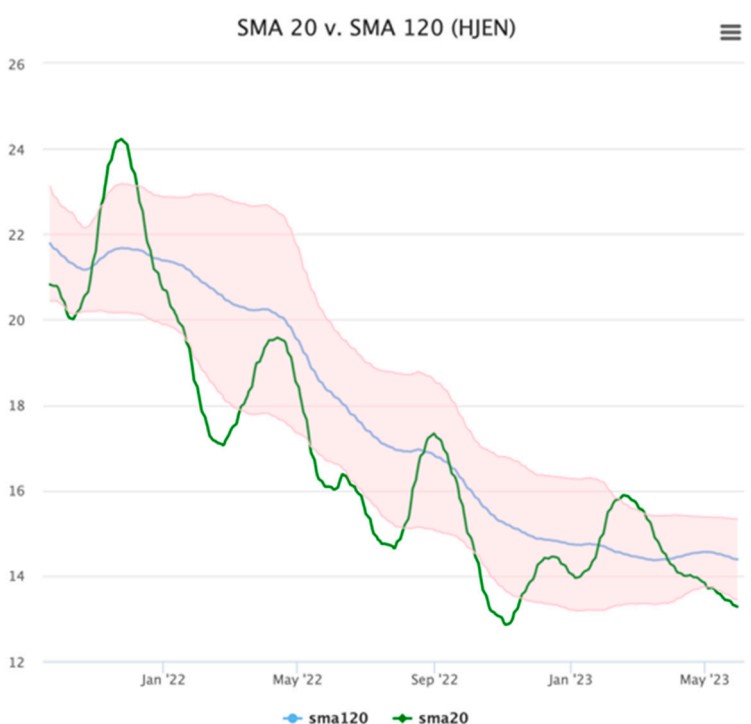

**Figure 7.** SMA20 versus SMA120 for HJEN.

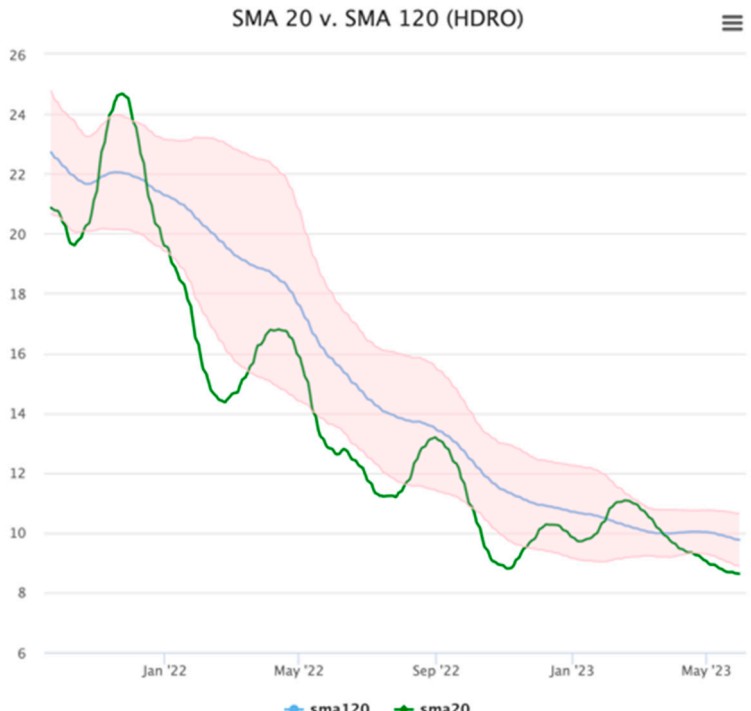

**Figure 8.** SMA20 versus SMA120 for HDRO.

For HDRO, there are two crossover points over the period from April 2021 to May 2023. These crossovers indicate shifts in the short-term and long-term trends, signaling opportunities for investors to adjust their positions based on prevailing market conditions. In the case of HJEN, there are three crossovers during the same period. These crossovers reflect more frequent changes in trend direction, offering investors additional chances to adapt their strategies.

Consequently, the Dual Moving Average Crossover (DMAC) strategies can serve as effective tools for preserving capital and minimizing losses in the green hydrogen sector. Figures 9 and 10 depict the DMAC strategy's equity curves versus the buy-and-hold approach for HJEN and HDRO, offering compelling insights into the effectiveness of this trading strategy for the two green hydrogen portfolios.

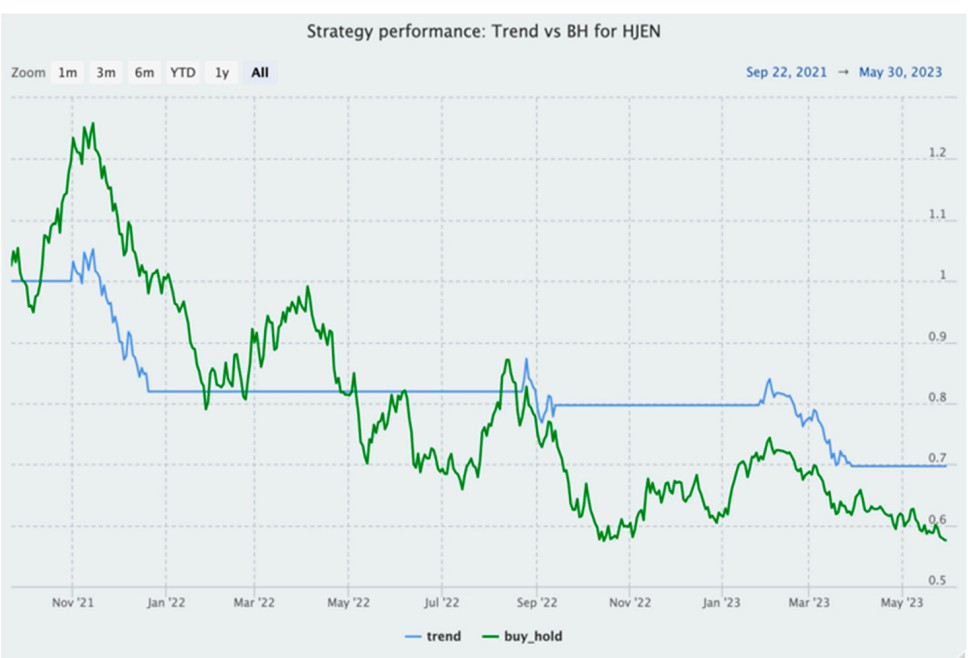

**Figure 9.** Equity curve of the DMAC trading strategy applied for HJEN.

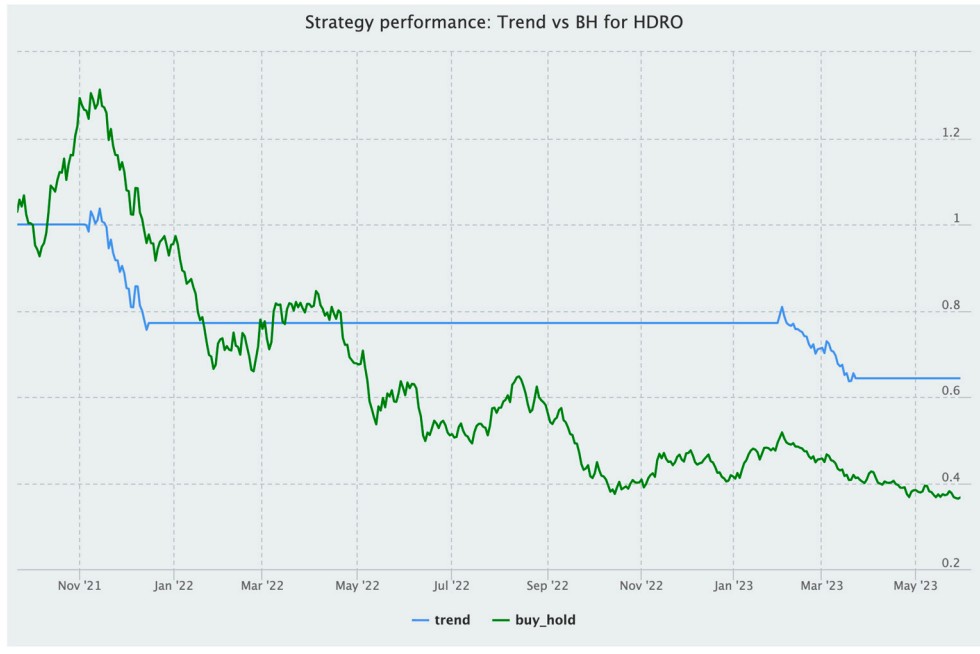

**Figure 10.** Equity curve of the DMAC trading strategy applied for HDRO.

In particular, the charts reveal that employing the DMAC strategy has led to improved performance, primarily by minimizing losses for both green hydrogen ETFs.

For HDRO, the equity curve under the DMAC strategy showcases a more favorable trajectory compared to the buy-and-hold curve. The value of a one-dollar investment

in HDRO increases from 0.38 to 0.62, indicating that the DMAC strategy has effectively reduced losses and preserved capital during market fluctuations. Similarly, for HJEN, the equity curve under the DMAC strategy exhibits a more positive evolution, with the value of one dollar growing from 0.58 to 0.7. This underscores the strategy's ability to minimize losses and enhance overall performance by identifying opportune entry and exit points.

These charts highlight the practical significance of employing systematic trading strategies, like DMAC, in the green hydrogen sector, where market conditions can be volatile and challenging to navigate. The ability of the DMAC strategy to reduce losses is particularly crucial for investors seeking to safeguard their capital while participating in this dynamic and specialized market. The visual representations emphasize that active strategies can mitigate downside risk and contribute to more favorable investment outcomes in the green hydrogen space. Overall, these findings underscore the crucial role of active trading strategies in dynamic markets that are subject to rapid shifts, supporting the conclusion of Dunis and Miao (2005).

In particular, the effectiveness of 20-120 Dual Moving Average Crossover (DMAC) trading strategies applied to green hydrogen ETFs is a critical consideration for investors in sustainable energy markets. Understanding the mechanism, rationale, and potential limitations of these strategies is essential for informed decision-making.

The 20-120 DMAC strategy involves two moving averages: a short-term 20-day moving average and a long-term 120-day moving average. The buy signal occurs when the 20-day moving average crosses above the 120-day moving average, indicating a potential upward trend, while a sell signal is generated when the 20-day moving average crosses below the 120-day moving average, suggesting a possible downward trend. The rationale behind this strategy is rooted in capturing trends specific to the green hydrogen market, allowing investors to capitalize on favorable price movements.

The green hydrogen market is dynamic (Biggins et al. 2022), influenced by factors such as technological advancements, policy changes, and global demand for sustainable energy solutions (Li et al. 2022; Lagioia et al. 2023). The 20-120 DMAC strategy aims to exploit trends arising from these factors. For instance, a bullish crossover could signal increased interest and positive sentiment toward green hydrogen, possibly driven by advancements in production technologies or favorable regulatory developments.

However, it's crucial to acknowledge potential limitations. The strategy may be sensitive to sudden market shifts or external shocks, as the green hydrogen sector is still evolving and subject to rapid changes. False signals and suboptimal performance might occur during periods of market uncertainty or when the ETF experiences heightened volatility. Moreover, the strategy assumes that historical price trends will continue, which may not hold in dynamic markets with emerging technologies.

## 3. Material and Methods

### 3.1. Data

The daily prices for five relevant ETFs from 1 April 2021 to 31 May 2023 are sourced from the Yahoo Finance platform, as per Table 3, using the getSymbols.yahoo() function in the {quantmod} package (Ryan et al. 2023) in R software version 2023.09.1+494. The analysis period is due to data availability, in particular, the inception dates for the two green hydrogen ETFs (i.e., 9 March 2021 for HDRO and 21 March 2021 for HJEN). Of note, all daily price series are subsequently transformed into logarithmic returns for modeling purposes. Table 3 depicts the research dataset.

**Table 3.** The research dataset.

| Ticker | Name | Exposure | Index Tracked | Source Location | Number of Observations |
|--------|------|----------|---------------|-----------------|------------------------|
| EFA | iShares MSCI EAFE ETF | large- and mid-capitalization developed market equities, excluding the U.S. and Canada, i.e., 900+ EAFE (Europe, Australia, Asia, and the Far East) companies | MSCI EAFE | https://finance.yahoo.com/quote/EFA/history?p=EFA, (accessed on 28 October 2023) | 543 |
| EEM | iShares MSCI Emerging Markets ETF | diversified large- and mid-capitalization emerging market equities (i.e., 800+ emerging market stocks) | MSCI Emerging Markets Index | https://finance.yahoo.com/quote/EEM/history?p=EEM, (accessed on 28 October 2023) | 543 |
| HJEN | Direxion Hydrogen ETF | 30 companies in the hydrogen industry, including hydrogen generation and storage, transportation and supply of hydrogen, fuel cells, and hydrogen fueling stations, with the US, UK, Japan, and France having the largest country weights | Hydrogen Economy Index (1H2ECO) | https://finance.yahoo.com/quote/HJEN/history?p=HJEN, (accessed on 28 October 2023) | 543 |
| HDRO | Defiance Next Gen H2 ETF | Globally listed companies that derive at least 50% of their revenue from hydrogen-based energy sources and fuel cell technologies (at least 25 constituents), with the US, south Korea, UK, and Canada having the largest country weights | BlueStar Hydrogen and NextGen Fuel Cell Index | https://finance.yahoo.com/quote/HDRO/history?p=HDRO, (accessed on 28 October 2023) | 543 |
| ICLN | iShares Global Clean Energy ETF | global equities in the clean energy sector, i.e., companies that produce energy from solar, wind, and other renewable sources | S&P Global Clean Energy Index | https://finance.yahoo.com/quote/ICLN/history?p=ICLN, (accessed on 28 October 2023) | 543 |

### 3.2. Visualization Techniques in R Software for Financial Data Analysis

The utilization of cumulative return charts, rolling performance charts, and SnailTrail charts in R software is instrumental in exploring and understanding the performance and behavior of the two green hydrogen portfolios. These visualization techniques enhance the analytical capabilities and provide a visual narrative of the data, aiding in decision-making and portfolio management.

Cumulative return charts are essential for understanding the overall performance of an asset or portfolio. The chart development implies calculating the cumulative returns of the two ETFs by accumulating the daily returns over the analysis time frame and subsequently using R's dedicated packages {PerformanceAnalytics} (Peterson et al. 2020) to create cumulative return charts that display the growth or decline of the portfolio's value over time, providing a visual representation of performance.

Rolling performance charts help assess the robustness of an investment strategy by analyzing performance over rolling windows. To create rolling performance charts for the two green hydrogen ETFs, the following steps were implemented:

- Define Rolling Windows: Specify the length of rolling windows that slide through the dataset. In this research, a 120-day rolling window is used to match the longer MA length in the Dual Moving Average Crossover (DMAC) strategy implementation.

- Calculate Performance Metrics: For each rolling window, calculate performance metrics such as returns, Sharpe ratios, or maximum drawdown.
- Visualization in R: Use R's {PerformanceAnalytics} package to generate line charts that display the performance metrics over the rolling windows with the charts.RollingPerformance() function. This helps identify trends and variations in performance throughout the historical data.

The SnailTrail chart is a novel visualization technique that traces the path of returns for each of the two green hydrogen portfolios, providing a dynamic view of their performance. To create a SnailTrail chart, the daily ETF returns are first used to calculate the cumulative return path by accumulating returns over time, thus creating a trajectory of portfolio growth, and then the specialized R chart.SnailTrail() function within {PerformanceAnalytics} is utilized to construct the corresponding SnailTrail charts. These charts offer an engaging way to observe the historical journey of the returns of the green hydrogen portfolios, offering further insights into their performance and thus assisting in informed decision-making.

### 3.3. Risk-Adjusted Performance Metrics and Downside Risk Analysis

The standard deviation Sharpe ratio (Sharpe 1998), which assesses returns relative to total risk, is an essential tool for measuring risk-adjusted returns. In this study, the standard deviation Sharpe ratio is computed as per Equation (1) by dividing the portfolio's average excess return $R_P$ (over a risk-free rate $r_f$ set to 0% in this research) by its standard deviation of returns $\sigma_P$, representing total risk.

$$\text{StdDev Sharpe} = \frac{R_P - r_f}{\sigma_P} \tag{1}$$

However, this metric can overlook the magnitude of downside risk. Consequently, the Expected Shortfall (ES) Sharpe and Value-at-Risk (VaR) Sharpe ratios given in Equations (2) and (3) address this limitation by incorporating the expected shortfall and value-at-risk metrics, respectively, which focus on the tails of the return distribution, emphasizing the impact of extreme losses.

ES Sharpe was thus determined by estimating the Expected Shortfall (or conditional VaR or tail VaR) (Tasche 2002; Nadarajah et al. 2014), which quantifies the expected loss in the tail of the return distribution beyond a certain threshold. ES Sharpe was calculated as the portfolio's average excess return divided by the ES as in Equation (2).

$$\text{ES Sharpe} = \frac{R_P - r_f}{ES_P} \tag{2}$$

Then, the VaR Sharpe was computed by estimating the Value-at-Risk (Stambaugh 1996; Linsmeier and Pearson 2000; Pflug 2000), representing the potential maximum loss at a specific confidence level (95% in this study). VaR Sharpe was calculated by dividing the portfolio's average excess return by the VaR as in Equation (3).

$$\text{VaR Sharpe} = \frac{R_P - r_f}{VaR_P} \tag{3}$$

These ratios provide different perspectives on risk-adjusted performance, with the standard Sharpe ratio focusing on volatility, ES Sharpe emphasizing the expected losses in the tail of the distribution, and VaR Sharpe considering the maximum potential loss at a specific confidence level.

The Sortino ratio (Sortino and Van Der Meer 1991; Sortino and Price 1994), on the other hand, specifically evaluates downside risk, providing a more nuanced perspective on portfolio performance. The Sortino ratio was calculated as in Equation (4) by taking the average excess return and dividing it by the downside standard deviation of returns. Downside deviation considers only returns below a specified target or threshold that

usually represents a minimum acceptable return (MAR) or the investor's downside risk tolerance (i.e., MAR is set to 0% in the study).

$$\text{Sortino ratio} = \frac{R_P - MAR}{\sigma_{MAR}} \tag{4}$$

where $\sigma_{MAR}$ is the downside deviation, given by Equation (5).

$$\sigma_{MAR} = \sqrt{\sum_{i=1}^{n} \frac{\min\left[(R_i - MAR), 0\right]^2}{n}} \tag{5}$$

Active Premium (Sharpe 1994), also dubbed 'active return', shows the return on an investment's annualized return minus the benchmark's annualized return, as per Equation (6).

$$\text{Active Premium} = \text{investment's annualized return} - \text{Benchmark's annualized return} \tag{6}$$

In turn, the Information ratio (Sharpe 1994) is further given by dividing the Active Premium by the Tracking Error, as in Equation (7), thus relating the degree to which an investment has beaten the benchmark to the consistency with which the investment has beaten the benchmark (Peterson et al. 2020):

$$\text{Information ratio} = \frac{\text{Active premium}}{\text{Tracking Error}} \tag{7}$$

The Treynor ratio is similar to the Sharpe Ratio, with the exception of employing the beta coefficient as the volatility measure, as in Equation (8):

$$\text{Treynor ratio} = \frac{R_P - r_f}{\beta_P} \tag{8}$$

Furthermore, the risk-return performance analysis is extended by comparatively assessing alternative portfolios using an array of downside risk metrics, including historical Value-at-Risk (VaR), modified VaR, Expected Shortfall (ES), loss deviation, downside deviation, and maximum drawdown, which in turn allow investors to evaluate performance beyond the scope of volatility and gain insights into a portfolio's resilience in turbulent market environments.

Value-at-Risk (VaR) is widely recognized as an industry standard for measuring downside risk in financial markets (Boudt et al. 2008; Shah et al. 2022; Hood and Malik 2018; Peterson et al. 2020). By providing a standardized and quantifiable measure of downside risk, VaR plays a critical role in risk management and regulatory compliance within the financial industry. The historical VaR at a probability level $p$ is the $p$-quantile of the negative returns, or equivalently, is the negative value of the $c = 1 - p$ quantile of the returns (Boudt et al. 2008; Peterson et al. 2020):

$$\text{VaR} = q.99 \tag{9}$$

where $q.99$ is the 99% empirical quantile of the negative return series.

Zangari (1996) and Favre and Galeano (2002) proposed a modified Cornish Fisher VaR calculation that takes the higher moments of non-normal distributions (skewness, kurtosis) into account using a Cornish Fisher expansion.

The Expected Shortfall (ES) (also called Expected Tail Loss (ETL) or Conditional Value at Risk (CVaR)) at a probability level p (e.g., 95%) is the negative value of the expected value of the return when the return is less than its $c = 1 - p$ quantile. The historical ES is thus estimated using the negative value of the sample average of all returns below the quantile (Boudt et al. 2008; Peterson et al. 2020).

The modified ES proposed by Boudt et al. (2008) takes the higher moments of non-normal distributions (skewness, kurtosis) into account using a Cornish-Fisher expansion (Peterson et al. 2020).

Both the ES and modified ES are estimated using the ETL() function within the {PerformanceAnalytics} package in R software by alternatively setting the "historical" and "modified" methods within the function specifications.

Finally, the maximum drawdown represents the most significant peak-to-trough decline in portfolio value over a specific period (Bacon 2023). It requires the estimation of the cumulative returns and the maximum cumulative returns (Peterson et al. 2020). When the cumulative returns are below the maximum cumulative returns, the series experiences a drawdown, estimated in this study with R's maxDrawdown() function in {PerformanceAnalytics}.

By considering these various metrics collectively, investors can make more informed decisions based on their specific risk tolerance and investment objectives, enhancing their ability to navigate unpredictable market conditions while prioritizing the preservation of capital.

### 3.4. Single-Factor Models

The study additionally employs single-factor models to offer further insights into the risk and return performance of the selected ETFs. Specifically, it calibrates the paramount Capital Asset Pricing Model (CAPM) developed by Sharpe (1964) and Lintner (1965) on daily HDRO and HJEN returns over the entire data sample. The CAPM assesses the systematic risk and expected returns of assets or portfolios based on their sensitivity to a single market factor–in this case, the developed markets index EFA is used for this purpose. The assessed model is given in Equation (5):

$$R_{it} - R_{ft} = \beta_{0i} + \beta_{1i}\left(R_{mt} - R_{ft}\right) + u_{it} \tag{10}$$

where $R_{it}$ denotes the ETF $i$ return and $R_{ft}$ is the rate of return for the risk-free asset so that the left-hand side of each equation represents the excess return of portfolio $i$, with $i$ alternatively standing for each of the green energy ETFs and for the emerging market portfolio (EEM). The right-hand side of the CAPM in Equation (1) includes the market factor excess return (i.e., $R_{mt}$ is proxied by the return of the developed markets ETF EFA), and $\beta_{1i}$ reflects the systematic risk characterizing the analyzed portfolios and the error term representing idiosyncratic risk.

### 3.5. Trend Trading Strategy Implementation

Dual Moving Average Crossover (DMAC) is a popular and straightforward technical analysis trading strategy that is widely used by traders and investors to identify trends and generate buy or sell signals in financial markets (Pätäri and Vilska 2014). The strategy involves two key components: the short-term moving average (e.g., 20-day) and the long-term moving average (e.g., 50-day, 120-day, or 200-day). By tracking the crossovers and divergences between these two moving averages, traders can make informed decisions on when to enter or exit positions. Buy and sell signals are generated based on crossovers of shorter-term SMAs (in this study, 20-day) over longer-term SMAs (here, 120-day). A crossover from below indicates a buy signal, while a crossover from above signals a sell. The trading rules and parameters are set consistently for all assets to maintain uniformity.

In this study, after trading signals were issued as per the above-described method, the Green Hydrogen portfolios were constructed by allocating funds consistent with the signals. Buy signals resulted in investments in the respective ETF, while sell signals led to divestments. The portfolio's initial capital, asset allocation, and any rebalancing rules were determined in advance. These decisions aimed to simulate real-world trading conditions and constraints. Finally, data visualization techniques were employed to assess the effectiveness of the trend-following strategies. To this end, performance comparisons were made between the DMAC-based portfolios and a benchmark proxied by the traditional "buy and hold" strategy on the same ETF.

The following items made up the trend strategy implementation in R software:

(1)    rolling 20-day SMA
(2)    rolling 120-day SMA
(3)    "If_else" logic to create a buy or sell signal:

- Buy when the Green hydrogen ETF (alternative HDRO and HJEN) 20-day SMA is above the 120-day SMA (a 'golden cross')
- Sell when the 20-day SMA moves below the 120-day SMA (a 'death cross').

(4)    HDRO buy-and-hold (BH) returns/HJEN BH returns
(5)    Signal-based trading on each ETF
(6)    Visualizing strategy performance versus buy-and-hold (BH)

Of note, the main goal of this analysis is not to identify the optimal SMA pairs for trading or investment strategies. Instead, the study employs one of the most widely used SMA pairs of 20-day and 120-day moving averages as a specific example for exemplification purposes, which in turn also suits the rather short lifespan of the green hydrogen investment funds. By focusing on this particular SMA combination, the research aims to provide a practical illustration of how SMAs can be utilized to improve the risk-return profile of green hydrogen investments. Thus, the choice of the SMA 20-120 pair is intended to serve as a straightforward and easily understandable model, allowing for clear and straightforward demonstrations of SMA-based trading signals and their application in decision-making processes. While other SMA combinations may offer varying degrees of effectiveness for specific strategies, the current exemplification merely intends to show the utility of the SMA 20-120 trading strategy in capturing price trends and minimizing exposure to adverse market movements.

All estimations throughout the study are performed in R software.

## 4. Conclusions

This research lies within the broader context of socially responsible investing (SRI). Sustainable finance, as exemplified by the principles of SRI, is founded on the dual pursuit of financial returns and positive societal impact. Previous literature suggests that SRI, which integrates ESG criteria into traditional financial analysis, has experienced mixed results regarding its impact on portfolio performance. While some studies have demonstrated the potential for higher risk-adjusted returns and positive performance in high-SRI portfolios, others have failed to establish a clear link between SRI and portfolio outcomes. This inconclusiveness underlines the need for continued research in this area to fully understand the potential benefits and trade-offs associated with SRI.

Green hydrogen, as a burgeoning and promising sector in the green energy domain, represents a pertinent yet under-investigated case study within this broader SRI framework. By providing a multifaceted analysis of the risk-return attributes of some relatively new investment vehicles, i.e., Green Hydrogen ETFs, compared with relevant conventional and green energy ETFs, the current study contributes to the evolving discourse on responsible investment strategies, offering new insights that are crucial for investors who wish to align their portfolios with sustainability goals. By evaluating various risk-return measures and downside risk statistics, this study equips investors, policymakers, and stakeholders with valuable insights to make informed decisions in their pursuit of sustainable and profitable investment strategies.

The main findings of the current research can be summarized as follows: (i) Green Hydrogen ETFs underperformed with lower returns and higher risk compared to conventional equity and green energy portfolios from April 2021 to May 2023. (ii) Single-factor models revealed that a substantial portion (i.e., more than 40%) of Green Hydrogen ETFs' returns were due to systematic market risk. (iii) Green Hydrogen ETFs showed high beta coefficients, negative alphas, and poor risk-adjusted metrics, indicating significant sensitivity to market fluctuations and an inability to reward investors after accounting for systematic risk. (iv) Implementing Dual Moving Average Crossover (DMAC) trading strategies significantly improved the risk-return performance of green hydrogen portfolios. This improvement

offers a pathway for investors to pursue sustainable and profitable investment strategies while aligning with financial and social objectives in their equity portfolios.

The multifaceted approach employed in this research, which combines a plethora of risk-adjusted performance metrics, downside risk analysis, single-factor models, and the implementation of trading strategies, enhances the depth of knowledge regarding Green Hydrogen ETFs' behavior. Furthermore, the study highlights the potential of Dual Moving Average Crossover (DMAC) trading strategies in enhancing performance in the green hydrogen sector, providing a critical advantage in the ever-changing landscape of this emerging market. It provides empirical evidence that adopting a systematic trading strategy like DMAC can be a valuable tool for investors navigating the complexities of the green hydrogen market. Consequently, the ability to minimize losses during market downturns underscores the importance of combining fundamental analysis with technical strategies in managing green hydrogen investments, ultimately contributing to a more resilient and potentially more profitable investment approach.

The strategies derived from this study offer valuable guidance for decision-makers navigating the dynamic landscape of green hydrogen investments. First and foremost, decision-makers should be cognizant of the inherent challenges in the green hydrogen sector, as evidenced by lower return performance and higher risk compared to conventional equity and green energy portfolios. In response to these challenges, the study advocates for a nuanced approach to portfolio evaluation, emphasizing the importance of employing an array of risk-adjusted performance metrics and downside risk statistics. By considering these measures in tandem, decision-makers can gain a holistic understanding of risk-return trade-offs, enabling more informed and tailored investment decisions. Additionally, the study underscores the significance of recognizing the exposure of Green Hydrogen ETFs to systematic market fluctuations, as revealed by single-factor models. Decision-makers are encouraged to factor in these sensitivities when formulating investment strategies, aligning expectations with the broader financial landscape. Furthermore, the study highlights the effectiveness of Dual Moving Average Crossover (DMAC) strategies in preserving capital and minimizing losses in the green hydrogen sector. Decision-makers can leverage these systematic strategies to navigate market downturns and foster resilience, potentially leading to more profitable investment approaches. In essence, the strategies derived from this study provide decision-makers with a comprehensive toolkit to navigate challenges, capitalize on opportunities, and stay responsive in the ever-changing market of green hydrogen investments.

Future research in the fields of socially responsible investing (SRI) and green hydrogen investments holds great promise for uncovering additional driving factors that impact portfolio performance. One direction for future inquiry involves the use of multifactor models to assess the influence of multiple variables beyond systematic risk, providing insights into how various elements interconnect and influence investment outcomes. Furthermore, research endeavors can explore the temporal dynamics of these multifactor models to assess how the importance of different factors may change over time. Not in the least, future research can benefit from an exploration of more intricate and sophisticated trading strategies. As markets continue to evolve, the development and testing of advanced trading algorithms, machine learning techniques, and artificial intelligence-driven strategies can offer novel insights and strategies for investors in the challenging green-hydrogen market.

**Funding:** This research received no external funding.

**Data Availability Statement:** All data is publicly available on the Yahoo Finance Platform.

**Conflicts of Interest:** The author declares no conflict of interest.

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
