# Peer review of "Enhancing Sustainable Finance through Green Hydrogen Equity Investments: A Multifaceted Risk-Return Analysis"

_risks, doi:10.3390/risks11120212_

Round 1
Reviewer 1 Report
Comments and Suggestions for Authors
1) While the introduction and literature review sections are comprehensive, I believe that particularly those studies that provide a critical perspective on green hydrogen investments could have been added.
2) The paper's reliance on the Capital Asset Pricing Model (CAPM) to assess risk and return may overlook multi-factor influences that are important in the volatile green energy space. Where possible, the integration of multi-factor models could provide a more nuanced understanding of ETF performance.
3) Based on the conclusions and discussion section, while providing valuable insights, I believe it could benefit from deeper analysis and interpretation, particularly in terms of the broader implications of the findings for the green hydrogen market and sustainable finance. In addition, the addition of a consideration of external factors would allow for a more thorough exploration of the impact of external factors such as regulatory changes, technological advances and geopolitical developments on green hydrogen investments, in order to place the findings within a broader economic and environmental framework.
Reviewer 2 Report
Comments and Suggestions for Authors
1. Clarify the gap, research question, and aim of the paper directly from the abstract.
2. Provide rationale for selecting DMAC and explain its appropriateness in your specific case.
3. It would be desirable if you could furnish information on the impact of Covid-19 on the energy market, particularly considering the substantial reduction in GHG emissions observed during the pandemic period.d.
4. The strategies derived from this study to guide decision-makers need clearer elucidation.
Reviewer 3 Report
Comments and Suggestions for Authors
After thoroughly reviewing the contents of the article titled "Enhancing Sustainable Finance through Green Hydrogen Equity Investments: A Multifaceted Risk-Return Analysis," it's evident that the analysis presented significant insights into the performance and risk attributes of Green Hydrogen Exchange-Traded Funds (ETFs) compared to conventional equity and green energy portfolios. The study effectively utilized data sourced from Yahoo Finance for relevant ETFs and conducted a comprehensive analysis from April 2021 to May 2023.
The paper assessed daily prices and logarithmic returns for five selected ETFs, depicting the evolution of daily closing prices over the specified period. It notably compared the performance and risk attributes, showcasing the underperformance of green hydrogen investments and their sensitivity to market fluctuations, as evidenced by high beta coefficients, negative alphas, and poor risk-adjusted metrics. Moreover, the article offered a silver lining by suggesting the implementation of straightforward Dual Moving Average Crossover (DMAC) trading strategies to enhance the risk-return performance of green hydrogen portfolios.
The discussion extensively covered the implications of the results, emphasizing the need for diversified portfolios, risk management strategies, and the necessity for investors to adapt to evolving market trends. The visual representations, such as figures and charts, provided a clear snapshot of the performance disparities among various assets, supporting the claims made in the analysis.
However, while the article provides a substantial understanding of the challenges and opportunities in the green hydrogen sector, it would benefit from improvements in several areas:
i) Results Interpretation:
The article could benefit from more in-depth interpretation of the results. It's crucial to not only present the findings but also to offer insights into the implications and significance of the observed trends within the broader financial context, and particularly in the green energy sector.
ii) Explanations about DMAC Strategies:
The discussion surrounding the effectiveness of Dual Moving Average Crossover (DMAC) trading strategies needs to be more detailed. Describing the mechanism and rationale behind these strategies and any potential limitations would add depth to the analysis.
By enhancing these specific aspects, the article's clarity, academic rigor, and overall contribution to the field of green energy investments and risk assessment would be significantly improved.
Reviewer 4 Report
Comments and Suggestions for Authors
The paper Enhancing Sustainable Finance through Green Hydrogen Equity Investments: A Multifaceted Risk-Return Analysis examines the risk and return of Green Hydrogen ETFs. Overall, the paper provides a novel and comprehensive analysis of an important future class of sustainable assets, and I see it as an interesting publication for the readership of the journal. I would advise the author to make several adjustments to the paper.
#1 Abstract requires restructuring. It is at the same time exhaustive (too much effort given on the explanation of the study background) and uninformative (it does not explain the method, results, and contributions of the study). The metadata of the study (title, abstract, keywords) are the first thing to be seen by the readership, and sometimes even the only thing read from this study, I would really like to see improvements in the quality of the abstract presentation.
#2 The fifth paragraph of the introduction includes both the aims, and the contributions of the study. It would be more beneficial to separate the two. Also, as early as in introduction (prior to contributions) the readership should get an indication of the findings.
#3 The datasets presented in Table 1 should go with a link. I could not find some of the sources.
#4 Methodology and the results are the stronghold of the paper. Great work!
#5 Conclusions require additional information. Bearing in mind that the paper does not have a discussion section, it would be beneficial to the readership to see the contextualization of the findings – in terms of comparing them to other similar studies. Also, the paper should have a clear statement on the limitations of the study.
#6 Some references are multiplied – see ref. no 5 and 6: Brzeszczynski, J., & McIntosh, G. (2014).; some references are not concise throughout the paper (see ref. no 36: Sortino) etc.
Comments on the Quality of English LanguageThe language requires minor adjustments (usually articles and other minor errors) to be in a camera-ready format, but overall the paper is of a high quality and easy to understand.
Round 2
Reviewer 1 Report
Comments and Suggestions for Authors
I have no problem with the author's revisions.
Author Response
Thank you for your input and positive feedback.
Reviewer 2 Report
Comments and Suggestions for Authors
All the comments were well addressed by the authors. However, I have a few additional remarks:
1. Please ensure that the legend in the graph of Figure 3 is more reader-friendly for improved readability.
2. The links in Table 1 are not functional. When clicked, they display an error from Yahoo.
3. Could you provide details on how the data used in the R package are updated? Is it necessary to update the entire software, or is updating the package alone sufficient?
Author Response
Dear Reviewer,
thank you for your feedback and cooperation. Please find below my answers:
1. I have redone Figure 3 to increase its readability by increasing the legend font and changing its location.
2. I have rechecked the links, and they seem to be correct. Please try to copy the links provided and then paste them into your browser's address bar to check their functionality and access the intended webpage.
3. In the 'quantmod' package in R, data updates can be managed independently of updating the entire software. The package incorporates functions like getSymbols() that fetch financial data from various sources like Yahoo Finance or other financial databases. By utilizing these functions, users can specifically update the data they need without updating the entire R software. Therefore, updating the 'quantmod' package is typically sufficient to access and retrieve the latest financial data without requiring a full software update.